# Enhancing E-Learning with Blockchain: Characteristics, Projects, and Emerging Trends

Mahmoud Bidry [1], Abdellah Ouaguid [2,3] and Mohamed Hanine [1,*]

1  LTI Laboratory, National School of Applied Sciences, Chouaib Doukkali University, El Jadida 24000, Morocco
2  RITM Laboratory, ESTC, Hassan II University, Casablanca 20000, Morocco
3  National Center for Scientific and Technical Research (CNRST), Rabat 10000, Morocco
*  Correspondence: hanine.m@ucd.ac.ma

**Abstract:** Blockchain represents a decentralized and distributed ledger technology, ensuring transparent and secure transaction recording across networks. This innovative technology offers several benefits, including increased security, trust, and transparency, making it suitable for a wide range of applications. In the last few years, there has been a growing interest in investigating the potential of Blockchain technology to enhance diverse fields, such as e-learning. In this research, we undertook a systematic literature review to explore the potential of Blockchain technology in enhancing the e-learning domain. Our research focused on four main questions: (1) What potential characteristics of Blockchain can contribute to enhancing e-learning? (2) What are the existing Blockchain projects dedicated to e-learning? (3) What are the limitations of existing projects? (4) What are the future trends in Blockchain-related research that will impact e-learning? The results showed that Blockchain technology has several characteristics that could benefit e-learning. We also discussed immutability, transparency, decentralization, security, and traceability. We also identified several existing Blockchain projects dedicated to e-learning and discussed their potential to revolutionize learning by providing more transparency, security, and effectiveness. However, our research also revealed many limitations and challenges that could be addressed to achieve Blockchain technology's potential in e-learning.

**Keywords:** Blockchain; distributed ledger; security; transparency; e-learning

## 1. Introduction

In recent times, Blockchain technology has garnered substantial attention owing to its transformative potential across various industries. The core of this technology revolves around the concept of Distributed Ledger Technology (DLT). DLT represents a decentralized system that enables numerous participants to collectively maintain and update a shared database [1]; by eliminating the necessity for a central authority, this approach enables greater transparency and data immutability. As a specific application of DLT, Blockchain stands out as the most popular and extensively explored in both practical applications and academic research [2]. Blockchain ensures secure transaction recording and verification over a network of connected computers by acting as a decentralized and transparent ledger. It operates on the principles of decentralization, immutability, consensus mechanisms, and cryptographic security [3,4].

The history of Blockchain technology traces back to the emergence of Bitcoin in 2008. Bitcoin, created by the pseudonymous figure known as Satoshi Nakamoto, introduced Blockchain as the underlying technology for cryptocurrencies [5]. However, the potential of Blockchain quickly transcended its initial use case. Over the years, Blockchain has evolved and found applications in various industries beyond finance [6,7]. Its secure and transparent nature has led to its exploration in sectors [8], such as supply chain management [7,9,10], healthcare [11,12], security analysis [13,14], open banking [15,16], e-learning [17,18], etc.

There are different types of Blockchains, each catering to specific needs and use cases.

Public Blockchains, exemplified by Bitcoin [5] and Ethereum [19], are accessible to anyone and emphasize a high degree of decentralization. The maintenance of these Blockchains relies on a distributed network of nodes, and anyone can participate in the network as a node, mine new blocks, and validate transactions [20,21].

Private Blockchains, on the other hand, are restricted to a specific group of participants and are often used by organizations for internal purposes. They provide a more controlled environment where access permissions are granted to authorized participants only. An illustration of a private Blockchain is Hyperledger Fabric, which is widely used in enterprise settings. It allows organizations to collaborate securely by sharing a private Blockchain network and maintaining control over who can join and participate in the network [20,21].

Consortium or permissioned Blockchains are governed by a consortium of organizations that have agreed upon a set of rules and access permissions [22]. These Blockchains integrate aspects from both public and private Blockchains. They offer a more collaborative approach where a predefined group of participants, typically from the same industry or consortium, work together to maintain the Blockchain network. An example of a consortium Blockchain is R3 Corda [23], which is used by multiple financial institutions to streamline processes such as trade finance and asset exchange [21,24].

The applications of Blockchain technology are extensive and continue to expand [7]. In the finance industry, Blockchain can facilitate secure and transparent transactions, eliminating the need for intermediaries and reducing costs. Supply chain management can benefit from Blockchain's ability to provide an immutable record of every transaction and ensure the authenticity of goods. In healthcare, Blockchain can enhance data interoperability, secure medical records, and enable better patient care coordination [12]. In the e-learning context, Blockchain technology ensures that all transactions and interactions are securely recorded on an immutable ledger. This transparency not only fosters a sense of accountability but also cultivates trust among learners, educators, and educational institutions.

Researchers acknowledge the disruptive potential of Blockchain technology in e-learning and have initiated research to explore its application in this domain. Our research aims to assess the progress of Blockchain research in e-learning, identify problems solved, and analyze the challenges faced when implementing Blockchain applications cases. To obtain it, we performed a literature review and conducted an empirical study of Blockchain in e-learning. According to our survey, using Preferred Reporting Items of Systematic Review and Meta-Analysis (PRISMA) guidelines [25], researchers in e-learning focus primarily on interactivity and platform issues and identify opportunities for privacy, security, data integrity, and transparency through usage [26]. We will also discuss the implications of various Blockchain technologies in e-learning, such as the potential for student engagement, student empowerment, and educational research development. In addition, we identify barriers and challenges faced in implementing Blockchain in e-learning and propose new research directions to overcome these barriers.

This review study will look into how Blockchain technology can improve the field of e-learning by examining its application to enhancing learning experiences. It will explore existing research, associated challenges, and possible avenues for further study. To steer our analysis, we have formulated the following research questions:

1.　RQ1: What potential characteristics of Blockchain can contribute to enhancing e-learning?
2.　RQ2: What existing Blockchain projects are dedicated to the field of e-learning?
3.　RQ3: What are the limitations of existing projects?
4.　RQ4: What are possible future research trends related to Blockchain that will impact the field of e-learning?

The remainder of the paper is organized as follows. In Section 2, we focus on related research and explore the limitations of the e-learning field, as well as exploring the features of Blockchain and its potential implementation in the e-learning domain. In Section 3, we present the research methodology. In Section 4, we present the findings obtained from addressing the research questions. In Section 5, we introduce the discussion. Lastly,

in Section 6, we summarize the limitations of our review and present potential future directions for research.

## 2. Literature Review

In this part of the paper, we will explain the e-learning field and its limitations, and how Blockchain technology can provide a solution for these limitations. We will examine the main features of the Blockchains and explore various studies related to Blockchain-based e-learning projects.

The e-learning field pertains to utilizing electronic technologies for delivering educational content and enabling remote learning. It encompasses various forms of online education, such as online courses, virtual classrooms, and interactive learning materials [27]. One of the key advantages of e-learning is its accessibility, as it allows individuals from diverse backgrounds to access education anytime and anywhere, overcoming geographical barriers. Additionally, e-learning offers flexibility in terms of pacing and scheduling, enabling learners to tailor their educational experience to their specific needs [28,29]. However, e-learning also poses certain technical limitations, particularly in the areas of data privacy and security [29]. As learners engage in online activities, their personal information and learning data are collected, raising concerns about privacy breaches and unauthorized access. Moreover, the reliance on a central platform in e-learning poses a significant risk. If the platform experiences a technical issue or shuts down, all the credentials and educational records stored within it may be lost [30].

Blockchain technology can play a vital role in addressing these challenges. It provides a range of features that enhance security and protect data privacy. The following key features of Blockchain contribute to its effectiveness in addressing these issues:

1.  **Decentralization:** Blockchain operates on a decentralized network of nodes, removing the need for a central authority. This decentralized architecture ensures that no single entity possesses complete control over the system, making it resistant to single points of failure and increasing overall system resilience [17,21,31].
2.  **Traceability:** The transparency and auditability of Blockchain enable the traceability of data and transactions. Each transaction recorded on the Blockchain is timestamped and linked to previous transactions, creating an immutable trail of information, which enhances accountability and trust [17,21].
3.  **Immutability:** Upon recording data on the Blockchain, it becomes nearly impossible to alter or delete. The immutability feature ensures the integrity and permanence of data, making it highly reliable for storing critical information [17,21,31,32].
4.  **Consensus Mechanism:** Blockchain utilizes consensus algorithms to validate and agree upon the state of the network. This distributed consensus guarantees that all participants in the network reach a mutual agreement on the validity of transactions, fostering trust and removing the necessity for intermediaries [17,21,31,33].
5.  **Smart Contracts:** Blockchain supports the execution of self-executing contracts known as smart contracts. These programmable contracts automatically enforce predefined conditions and actions, eliminating the need for intermediaries and reducing transaction costs while increasing efficiency and transparency [19,21,34].

By leveraging these features, Blockchain technology establishes a robust and reliable foundation for tackling the privacy and security of the education field [35]. One of the key advantages of Blockchain is its ability to empower users with greater control over their data. Through the use of private keys and cryptographic techniques [31], learners can maintain ownership of their educational records and decide who can access their information. This puts individuals in charge of their own data, reducing the reliance on centralized platforms and minimizing the risks associated with data breaches and unauthorized access.

Moreover, Blockchain enhances trust within the e-learning ecosystem. The decentralized and transparent nature of Blockchain ensures that all transactions and interactions are recorded on an immutable ledger, visible to all network participants. This transparency fosters accountability and builds trust among learners, educators, and institutions, as the

integrity of the system is verifiable and tamper-resistant. Learners can have confidence in the authenticity of educational resources, credentials, and certifications [36], as these can be traced back to their source and validated through the Blockchain's decentralized consensus mechanisms.

## 3. Methodology

In this section, we outline the methodological approach employed for our systematic review. Divided into two subsections, namely "Search Process and Selection" and "Data Extraction", our methodology adheres to established guidelines, ensuring a structured and transparent review. By integrating the PRISMA framework, we ensure methodical data synthesis. We detail our search strategy, study selection criteria, and systematic data extraction, offering insights into the systematic foundation of our review.

### 3.1. Search Process and Selection

In this scientific review, we conducted an in-depth search of various scientific articles that touch on the field of Blockchain and e-learning to answer our research questions. We used the PRISMA approach to identify, select, evaluate, and synthesize relevant studies for our research questions. By adopting the PRISMA approach, we can showcase the review's quality, assist readers in assessing its strengths and weaknesses, enable replication of review methods, and organize the review using PRISMA headings [37,38]. We used keywords such as "Blockchain", "E-learning", "Education", "Decentralization", "Security", and "Privacy" to select relevant articles. We examined titles, abstracts, and keywords to assess the relevance of each article. Inclusion criteria were that articles had to be in English, published between 2016 and 2023, and specifically address the use of Blockchain in e-learning. Exclusion criteria were articles that had no direct link to our research topic.

### 3.2. Data Extraction

In the 'Identification' phase, the PRISMA approach (Preferred Reporting Items for Systematic Reviews and Meta-Analyses) in Figure 1 was used in our paper selection process. We aimed to gather relevant and reliable research by searching through reputable sources, namely ScienceDirect, Scopus, and Google Scholar. This initial search yielded a total of 710 papers (n = 710).

Moving on to the 'Screening' phase, we took steps to eliminate duplicate papers to avoid redundancy and ensure the uniqueness of our review. After removing duplicates, the number of papers decreased to 567 (n = 567). We then applied additional exclusion criteria to further refine our selection. We excluded papers that were not directly related to our specific topic, resulting in the elimination of 291 papers (n = 291). Furthermore, papers that were not written in English were excluded, as well as those published before 2016. Additionally, we excluded editorial articles, opinion pieces, and other non-research articles. This screening process allowed us to focus on papers that were more closely aligned with our research objectives.

In the subsequent 'Eligibility' phase, we further refined our paper selection by carefully examining the remaining papers for their relevance to our research question, aims, and objectives. Through this meticulous assessment, we excluded an additional 229 papers (n = 229) that did not meet our criteria. As a result, we arrived at a final count of 47 papers (n = 47) that were directly pertinent to our study.

By strictly adhering to the PRISMA approach throughout the review process, including the application of these exclusion criteria in the 'Identification' and 'Screening' phases, we ensured the inclusion of these 47 pertinent papers. This rigorous methodology enhances the reliability and validity of our findings, as it minimizes bias and ensures that the selected papers align with the objectives and scope of our research.

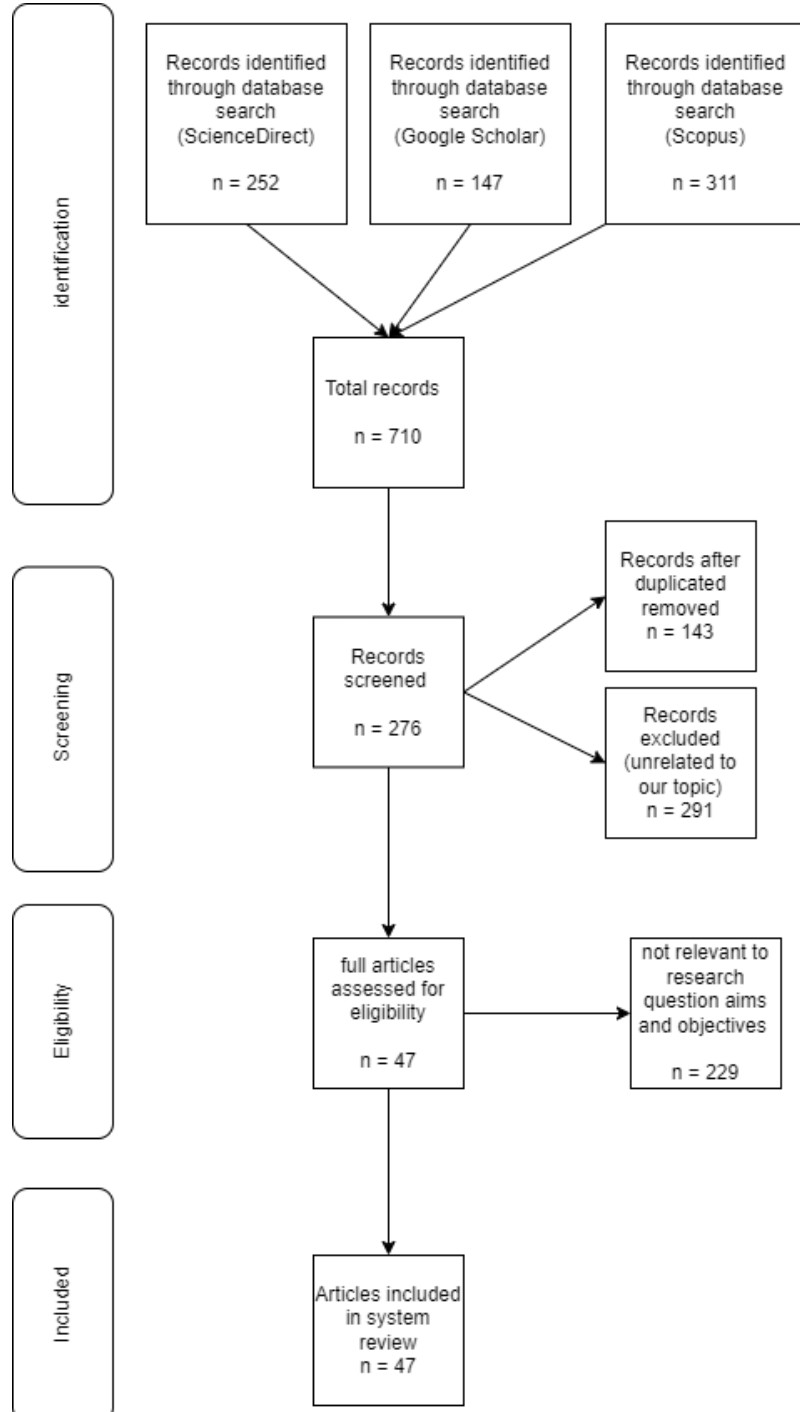

**Figure 1.** PRISMA diagram illustrating our literature research methodology.

The pie graph depicted in Figure 2 represents the distribution of three different databases used in the systematic review. These databases include ScienceDirect, Scopus, and Google Scholar. Each database's contribution is depicted as a percentage of the total. The largest portion of the pie is occupied by Scopus, which accounts for 43.8% of the data used in the system review. This indicates that Scopus played a significant role in providing information and resources for the study. Following closely behind is ScienceDirect, representing 35.5% of the data. ScienceDirect also made a substantial contribution to the system review, though slightly less than Scopus. Lastly, Google Scholar makes up 20.7% of the pie, indicating a relatively smaller but still notable portion. Google Scholar's database

was utilized to a lesser extent compared to the other two, but it still played a meaningful role in the overall study.

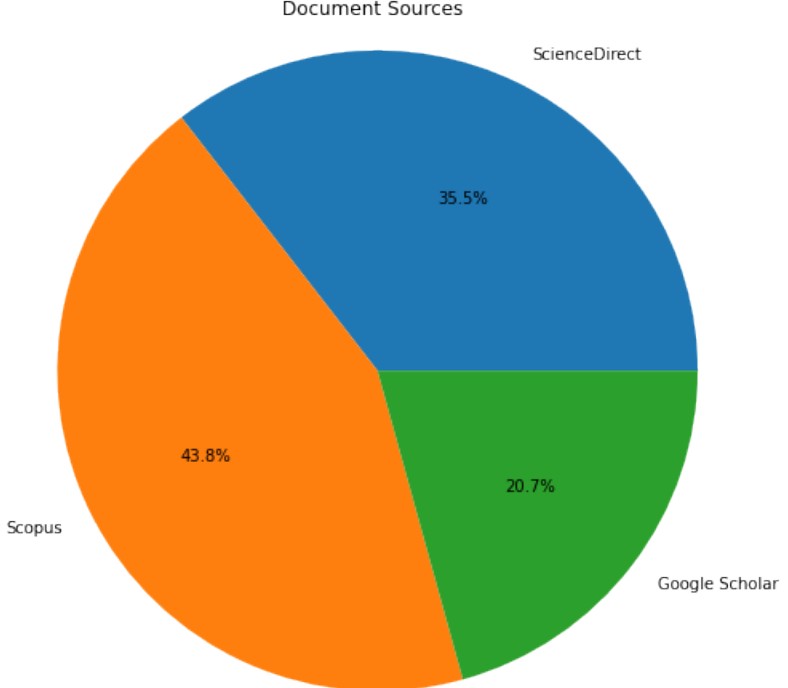

**Figure 2.** Document sources.

The graph in Figure 3 shows the contribution of each bibliographic database to the total pool of papers. The number of papers recovered from each database is shown in the blue bar on the left, both in terms of the total number of papers retrieved and as a percentage of those papers. The number of pertinent papers, included and kept, is shown in the orange bar on the right. Again, this number is provided in absolute numbers and as a percentage of all the papers retrieved from that particular database.

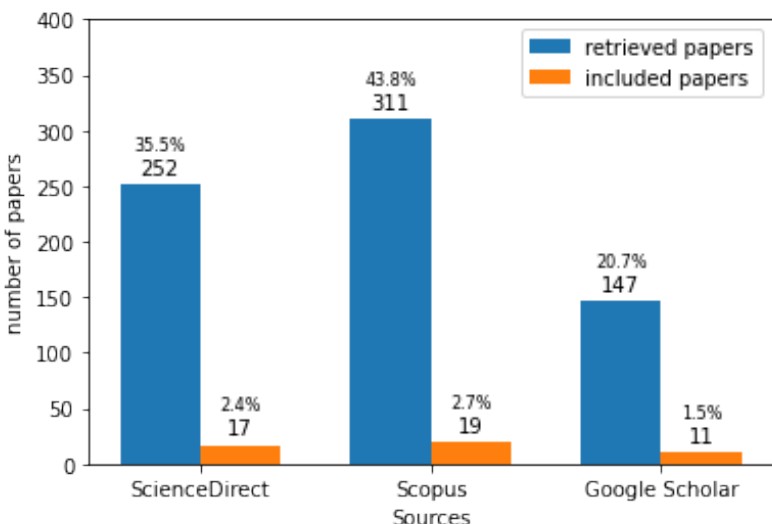

**Figure 3.** Retrieved and included papers.

The pie graph in Figure 4 titled "Document Types" illustrates the distribution of document types used in the study. Five distinct classes are represented in the graph, each corresponding to a specific percentage. The largest portion of the pie is occupied by research

articles, accounting for 41.4% of the total. Review articles make up 13.7% of the document types, while book chapters represent a smaller percentage at 6.8%. Conference papers contribute significantly to the study, comprising 24.4% of the pie. The remaining 13.8% falls under the category of "Other", which encompasses various miscellaneous document types.

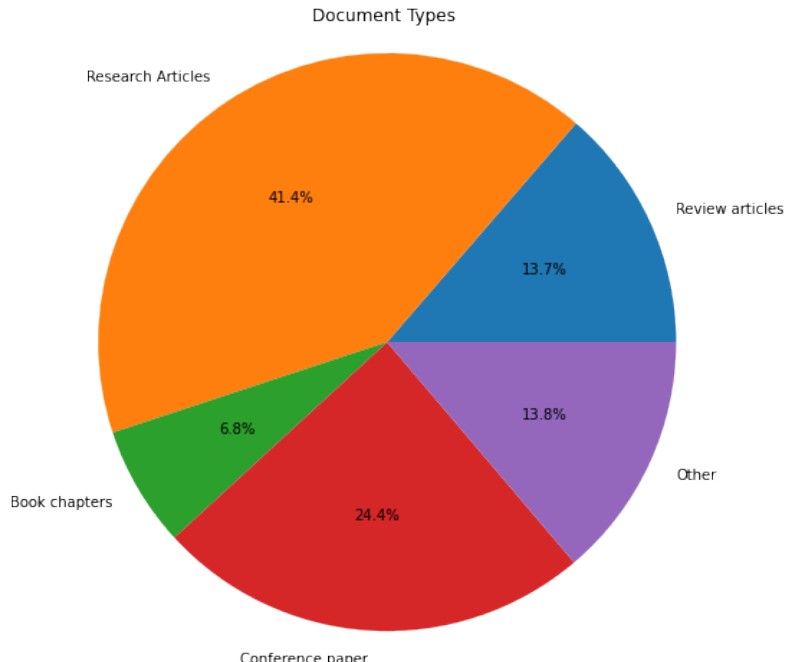

**Figure 4.** Document types.

In the analysis of the research, two graphical representations are utilized: a word cloud graph (Figure 5) and a bar chart graph (Figure 6). These visuals offer valuable insights into the keywords used throughout the research.

The word cloud graph presents a visually striking representation of the keywords. The size and prominence of each word within the cloud reflect its frequency of occurrence in the research. Larger and more prominent words indicate keywords that are mentioned more frequently, such as "Blockchain", "Education", and "Learning". By observing the word cloud, one can readily identify the most significant and prevalent topics or themes covered in the research.

Complementing the word cloud, the bar chart graph in Figure 6 focuses on the occurrence of the top keywords. Each bar on the chart corresponds to a specific keyword, and its length indicates the frequency with which the keyword appears in the research. This representation enables a direct comparison of the relative occurrences of different keywords, providing insights into the key areas of focus or emphasis in the research.

Together, these visualizations enhance our understanding of the research by capturing the frequency and prominence of keywords. They allow for quick identification of the most prevalent themes and provide valuable insights into the primary areas of concentration within the research.

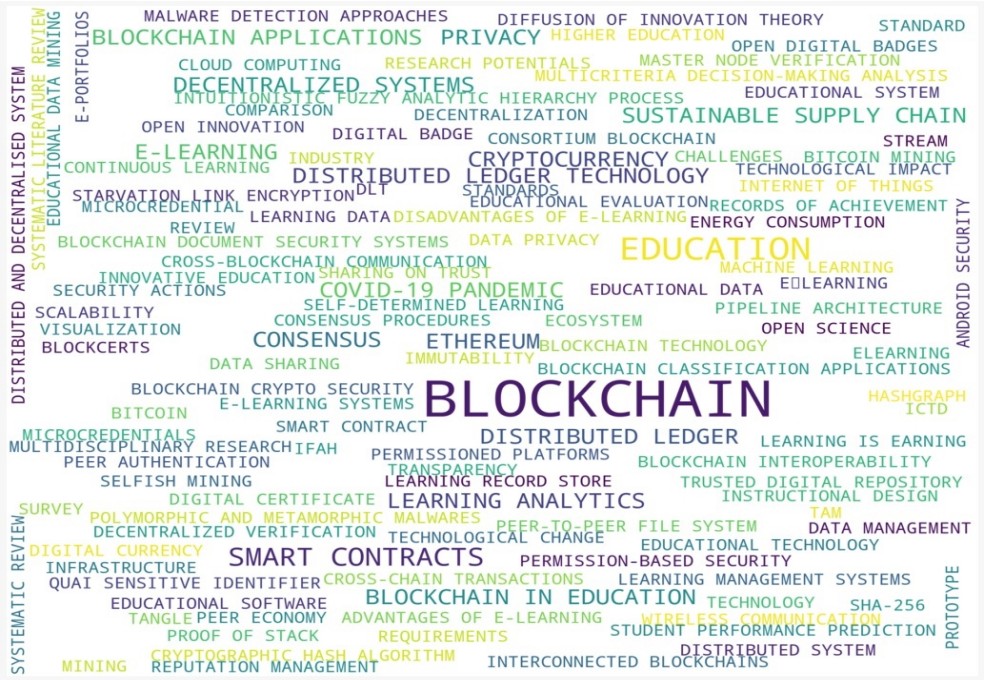

**Figure 5.** Word cloud of the most frequently used keywords in this research.

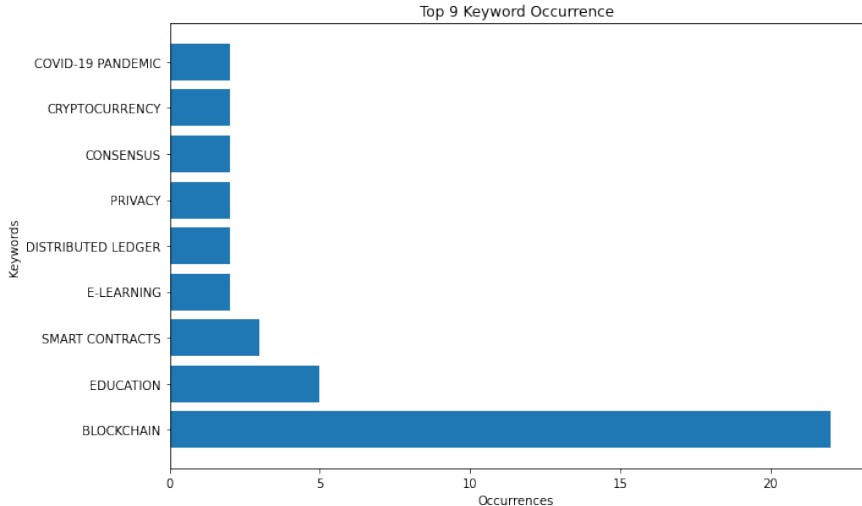

**Figure 6.** Bar chart graph of the most frequently used keywords in this research.

## 4. Results

In this section, we provide a comprehensive analysis of our findings in response to the four central questions guiding this systematic review. We address the potential attributes of Blockchain technology that hold promise for elevating e-learning experiences, highlighting their significance within this educational context. Additionally, we delve into an examination of existing dedicated Blockchain projects in the field of e-learning, aiming to uncover their contributions and implications. Notably, our exploration extends to the identification and elucidation of limitations inherent in these projects, offering insights into the challenges they may encounter. Lastly, we embark on a forward-looking trajectory, projecting future trends in Blockchain-related research that are poised to shape and transform the landscape of e-learning.

### 4.1. The Characteristics of Blockchain Can Enhance the Field of E-Learning

Blockchain technology has significantly improved e-learning processes, revolutionizing education delivery, quality assessment, and transactional maintenance. It provides

unique features such as immutability, transparency, decentralization, security, and traceability, backed by smart contracts and safe cryptocurrency exchange options. Implementing Blockchain enhances data integrity and verification processes, leading to increased trust in educational channels for students, educators, and employers.

Immutability and transparency, as highlighted in [39–42], are key features of Blockchain technology that enable the secure storage of learners' data [43], including interactions, assessments, feedback, and achievements. These features benefit learners by allowing them to track their progress, educators by allowing them to monitor student performance, and employers by allowing them to verify skills and competencies [44]. Decentralization and security, also emphasized in [39–41], ensure the authenticity and validity of learners' certificates and credentials [43] across institutions and platforms, enabling learners to prove their qualifications, educators to prevent fraud and forgery, and employers to reduce hiring costs and risks. The traceability and provenance features, mentioned in [39,40,42], safeguard the intellectual property rights of course creators and providers, ensuring content integrity and preventing plagiarism. This benefits learners by providing access to high-quality and original content, educators by enabling them to monetize their work and reputation, and providers by enforcing terms of use. Smart contracts and cryptocurrencies, as discussed in [39–41], streamline payments for e-learning services and products using digital tokens, enabling learners to make cost-effective and fast transactions, educators to receive fair compensation without intermediaries, and providers to offer flexible pricing models based on demand or supply.

### 4.2. Exploring Blockchain Projects in E-Learning

Blockchain-powered education is experiencing a lot of innovation. Many e-learning platforms are being developed, such as IEEE Blockchain eLearning Modules [45], Interactive Learning Experience Platform (ILEP) [42], APPII [21], ODEM [21], Blockcerts [46], etc. These platforms have different features, such as verifying certificates and tracking student performance. They also provide secure data sharing. These educational platforms are expected to revolutionize learning in different fields by offering more transparency, security, and effectiveness.

Here are some additional details about each platform:

1. **IEEE Blockchain eLearning Modules:** This is a series of online e-learning modules on Blockchain offered by the IEEE Blockchain Initiative. The modules cover topics such as Blockchain fundamentals, applications, challenges, and opportunities. The modules are designed to assist learners in comprehending how Blockchain can provide a novel approach to conducting transactions, ensuring network security, and recording the authenticity and source of data [45].
2. **Interactive Learning Experience Platform (ILEP):** This e-learning platform is a proof of concept developed by researchers from University College London, leveraging Blockchain technology. The primary objectives of the platform are to enhance transparency in assessments and enable personalized curriculum delivery within the higher education context. The platform can automate assessments and issue credentials using Blockchain technology [42].
3. **APPII:** This platform uses the Ethereum Blockchain to allow users to create their profiles and smart CVs with verified courses from various e-learning platforms. Users can also earn tokens for completing courses or verifying other users' credentials. The platform uses smart contracts to ensure that users' data are secure and verifiable [21].
4. **ODEM:** This platform uses the Ethereum Blockchain to connect students with educators using smart contracts. Students can browse and book courses from various providers, pay with cryptocurrencies, and receive certificates on the Blockchain. Educators can create and offer courses, set their own prices, and receive payments directly [21,47].

5.  **Blockcerts:** The public Blockchain Blockcert project of MIT [3]. This platform uses Bitcoin Blockchain to create, issue, view, and verify Blockchain-based certificates. The platform uses open standards and protocols to ensure interoperability and compatibility across different platforms and institutions. The platform also allows users to store their certificates in a digital wallet that they control [21,43,46].

6.  **Sony Global Education (SGE):** Sony Global Education (SGE) has pioneered technology that applies Blockchain to the educational domain, utilizing its secure properties to enable encrypted data transmissions. This includes academic proficiency records and progress measurements, facilitating secure exchanges between specific parties [48]. This platform is renowned for safeguarding and sharing student records based on Blockchain technology [21,49].

7.  **University of Nicosia and Blockchain:** The University of Nicosia (UNIC) stands as the sole university offering comprehensive Blockchain credentials [46]. UNIC utilizes Blockchain technology to manage students' records, including certificates they have obtained from MOOC platforms [17,50].

*4.3. Limitations of Existing Projects*

While Blockchain technology has the potential to bring many benefits to the field of e-learning, there are also some limitations and challenges that need to be addressed. Some of the challenges and limitations associated with employing Blockchain technology in e-learning include:

1.  **Scalability issue:** Among the primary challenges of using Blockchain technology in e-learning is its ability to scale to meet the demands of large-scale systems. Blockchain networks can become congested as the number of users and transactions increases, leading to slower processing times and higher transaction fees. This can be a significant challenge for e-learning systems that require support for a vast number of users and transactions [8,51,52].

2.  **Immutable issue:** In e-learning, errors in content or assessments may occur, requiring corrections or updates. However, once data are recorded on a Blockchain, it becomes difficult to make changes. This can be problematic if incorrect information or outdated content is permanently stored on the Blockchain, affecting the learning experience. Balancing the advantages of immutability with the need for error correction and content updates is a challenge that needs to be addressed [53].

3.  **Interoperability issue:** Another challenge of using Blockchain technology in e-learning is ensuring interoperability between different systems. There are many different Blockchain platforms and standards, and it can be difficult to ensure that data and transactions can be seamlessly shared between different systems. This can be a significant challenge for e-learning systems that need to integrate with other systems and platforms [54].

4.  **Regulation issue:** The use of Blockchain technology in e-learning may also be subject to various legal and regulatory requirements. For example, there may be requirements around data protection, privacy, and security that need to be considered when using Blockchain technology in e-learning [42]. These requirements can vary between different jurisdictions, which can make it challenging for e-learning systems to comply with all relevant regulations [51].

5.  **Cost and Performance issue:** Implementing Blockchain solutions can be expensive, both in terms of infrastructure requirements and energy consumption [55]. As e-learning platforms often have a vast amount of data to store and process, the costs associated with Blockchain transactions and storage can be a significant challenge [42]. Additionally, the time required for transaction confirmation and block validation may not meet the real-time demands of e-learning applications [55]. Furthermore, the cost of transactions in Blockchain networks can also be a significant challenge [56]. The high transaction fees associated with some Blockchain networks may make it difficult for e-learning platforms to adopt this technology in a cost-effective manner.

Education budgets vary widely between countries and regions, and it is important for governments and educational institutions to carefully evaluate the costs and benefits of implementing Blockchain technology and allocate appropriate resources to support its adoption [57].

6. **User Experience issue:** Blockchain technology can present a complex user experience, requiring learners and instructors to understand concepts such as wallets, private keys, and transaction processes [42]. For e-learning platforms aiming to cater to a broad range of users, particularly those with limited technical expertise, the complexity of interacting with Blockchain can be a barrier to adoption and usability.

Let us now draw a comprehensive table (see Table 1) encompassing all the aforementioned projects, accompanied by their respective descriptions and key features.

**Table 1.** Existing projects with their key features.

| Project | Description | Feature |
|---|---|---|
| IEEE Blockchain eLearning Modules | A series of online e-learning modules on Blockchain offered by the IEEE Blockchain Initiative. The modules cover topics such as Blockchain fundamentals, applications, challenges, and opportunities. | - Online e-learning modules<br>- Covers a range of topics related to Blockchain<br>- Offered by the IEEE Blockchain Initiative |
| Interactive Learning Experience Platform (ILEP) | This e-learning platform is a proof of concept developed by researchers from University College London, leveraging Blockchain technology. The primary objectives of the platform are to enhance transparency in assessments and enable personalized curriculum delivery within the higher education context. | - Blockchain-based e-learning platform<br>- Increases transparency in assessments<br>- Facilitates curriculum personalization |
| APPII | A platform that uses the Ethereum Blockchain to allow users to create their profiles and smart CVs with verified courses from various e-learning platforms. Users can also earn tokens for completing courses or verifying other users' credentials. | - Uses Ethereum Blockchain<br>- Allows users to create their profile and smart CVs<br>- Users can earn tokens for completing courses or verifying other users' credentials |
| ODEM | A platform that uses the Ethereum Blockchain to connect students with educators using smart contracts. Students can browse and book courses from various providers, pay with cryptocurrencies, and receive certificates on the Blockchain. | - Uses Ethereum Blockchain<br>- Connects students with educators using smart contracts<br>- Students can browse and book courses from various providers<br>- Students can pay with cryptocurrencies and receive certificates on the Blockchain |
| Blockcerts | A public Blockchain project of MIT that uses the Bitcoin Blockchain to create, issue, view, and verify Blockchain-based certificates. The platform uses open standards and protocols to ensure interoperability and compatibility across different platforms and institutions. | - Uses Bitcoin Blockchain<br>- Creates, issues, views, and verifies Blockchain-based certificates<br>- Uses open standards and protocols to ensure interoperability and compatibility |
| Sony Global Education (SGE) | Sony Global Education (SGE) has pioneered technology that applies Blockchain to the educational domain, utilizing its secure properties to enable encrypted data transmissions. This includes academic proficiency records and progress measurements, facilitating secure exchanges between specific parties. | - Applies Blockchain to the educational field<br>- Realizes encrypted transmission of data such as academic proficiency records and measures of progress |
| The University of Nicosia and Blockchain | The University of Nicosia (UNIC) stands as the sole university offering comprehensive Blockchain credentials [46]. UNIC utilizes Blockchain technology to manage students' records, including certificates they have obtained from MOOC platforms. | - Provides full Blockchain credentials<br>- Uses Blockchain technology to manage student records |

*4.4. Possible Future Research Trends Related to Blockchain Will Impact the Field of E-Learning*

Blockchain technology holds significant potential for transforming the field of e-learning. As technology continues to advance, several future research trends are anticipated to exert a substantial influence on the convergence of Blockchain and e-learning. Various scholarly works have proposed potential research directions in this area.

The authors of [42] suggested that Blockchain technology has the potential to enhance transparency and foster trust in assessment processes and educational credentials. There is growing interest in using Blockchain technology in e-learning, with several projects already underway. However, this is still a relatively new area of research and development, so there may be challenges or limitations to implementing Blockchain-enabled e-learning platforms that have yet to be fully explored. Ref. [58] suggested that Blockchain can enhance privacy and security in academic records while also providing a more efficient way to share information between institutions. Blockchain can also help to reduce fraud in academic credentials and provide a more transparent way of tracking student progress. However, it is important to note that there are still challenges facing the implementation of Blockchain in education that need to be addressed. According to [59], Blockchain technology can be applied in various ways within the field of education, such as digital credentialing and verification, secure peer-to-peer transactions, and educational administration tasks. Ref. [3] highlighted the benefits of Blockchain technology for smart learning environments, including collaborative learning, privacy preservation, and enhanced trust and transparency in educational transactions and records. Ref. [50] emphasized the potential benefits of using Blockchain technology in online learning platforms, such as increased transparency, security, and efficiency, while [60] suggested various potential applications in educational contexts, such as decentralized verification of academic credentials and secure storage of student records. Ref. [61] mentions the benefits of using a Blockchain system for storing university grades, including tamper-proof and transparent records and automatic allocation of awards via smart contracts. Finally, Ref. [62] mentions that the metaverse and Blockchain can be used in education to enhance the learning experience and provide greater access to education. The metaverse can provide an immersive and customizable learning environment that can improve concentration, understanding, and retention. It can also be used for interviews and preliminary assessments, which can give a great boost and improvement to training phases. The Blockchain can provide a secure and transparent way to store and share educational records, certifications, and achievements, reducing fraud and increasing trust in the education system. The combination of the metaverse and Blockchain can create a powerful tool for education that is secure, transparent, and accessible to all.

## 5. Discussion

Our research focused on enhancing the e-learning through advanced technologies such as Blockchains. The findings highlight numerous advantages that Blockchain offers, including increased security, reliability, and transparency in educational systems. We evaluated various dedicated Blockchain projects specifically developed to revolutionize e-learning processes, such as IEEE Blockchain eLearning Modules, Interactive Learning Experience Platform (ILEP), APPII, and ODEM Blockcerts. These platforms aim to enhance different aspects of learning by providing trustworthy systems based on transparency, safety, and efficiency. They also enable secure data sharing for student attendance verification and performance tracking. However, our assessments also uncovered limitations, including scalability issues, immutability challenges, high operational expenses, and emerging regulatory obstacles, which hinder progress and user experiences. Despite these limitations and challenges, our study revealed significant potential benefits and opportunities associated with Blockchain technology in e-learning. For instance, further research can focus on leveraging Blockchain for secure identification (IDs), digital certificates, and badges to enhance student appeal in the job market. Furthermore, the possibility that this key technology could be used for facilitating student empowerment by enabling peer-to-peer learning and collaborations, as well as improving accessibility and affordability of ed-

ucation. By exploring these avenues, we can unlock the full potential of Blockchain in e-learning and shape the future of education. Our study has its own limitations as it focused on specific research questions and may not have explored the full range of applications and challenges of Blockchain technology in e-learning. Additionally, the dynamic nature of Blockchain technology means that new advances may have emerged since our research was completed that require constant exploration and adaptation. Furthermore, it is important to acknowledge the limitations related to the search process employed in our study. We restricted our search to articles indexed in ScienceDirect, Scopus, and Google Scholar, within the timeframe of 2016 to 2023. While these databases are widely recognized, they may not capture all relevant publications on the topic. Additionally, our focus on English language papers introduces a potential language bias and overlooks contributions from non-English sources. Considering these limitations, it is crucial to interpret the findings of this review within the context of these restrictions. To enable a more thorough analysis of the integration of Blockchain technology in the learning sector, future research should try to lengthen the search period, include a wider range of databases, and take publications in several languages into account. By addressing these limitations and conducting more inclusive research, we can have a better understanding of how Blockchain technology can affect online education.

## 6. Conclusions

In conclusion, our study has provided valuable insight into the potential of Blockchain technology to revolutionize e-learning. Through our research, we found that Blockchain offers several beneficial properties, including increased security, trust, and transparency, which can significantly benefit e-learning systems. We have also identified specific Blockchain projects dedicated to e-learning that promise to transform the way we learn by providing more transparent, secure, and efficient learning experiences. However, it is important to recognize the limitations and challenges that remain to be addressed in order to fully harness the potential of Blockchain in e-learning. Our study has its own limitations, as it focused on specific research questions and may not have explored the full range of applications and challenges of Blockchain technology in e-learning. Additionally, the dynamic nature of Blockchain technology means that new advances may have emerged since our research was completed that require constant exploration and adaptation. Looking ahead, there are promising perspectives for integrating Blockchain technology into e-learning environments. As Blockchain continues to evolve and mature, we anticipate further advancements that could fundamentally change the educational experience. Blockchain has the potential to empower students through peer-to-peer collaboration, personalized learning paths, and secure authentication systems. It can also improve accessibility and affordability, enabling students from diverse backgrounds to access quality education.

**Author Contributions:** Conceptualization, M.B. and M.H.; methodology, M.B., A.O. and M.H.; validation, A.O. and M.H.; formal analysis, M.B. and A.O.; investigation, A.O.; resources, M.H.; data curation, M.B.; writing—original draft preparation, M.B. and A.O.; writing—review and editing, M.B. and A.O.; visualization, M.B., A.O. and M.H.; supervision, M.H. and A.O.; project administration, M.H. All authors have read and agreed to the published version of the manuscript.

**Funding:** This research received no external funding.

**Data Availability Statement:** The data presented in this study are available on request from the corresponding author.

**Conflicts of Interest:** The authors declare no conflict of interest.

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
