# Peer review of "Enhancing E-Learning with Blockchain: Characteristics, Projects, and Emerging Trends"

_futureinternet, doi:10.3390/fi15090293_

Round 1
Reviewer 1 Report
I congratulate the authors on an interesting and timely review, which I read with pleasure. However, there is one problem in blockchain technologies, which is not considered in the review, but can be the main limiter of their applications in education - this is the cost of transactions, which is still quite high in all blockchain networks. It seems to me that comparing the cost of blockchain platforms with traditional approaches in distance education should be an important addition to the review.
Reviewer 2 Report
Authors try to provide characteristics, projects, and emerging of the potential of blockchain technology to enhance e-learning, focusing in four main questions: (1) What potential characteristics of Blockchain can contribute to enhancing e-learning? (2) What are the existing Blockchain projects dedicated to e-learning? (3) What are the limitations of existing projects? (4) What are the future trends in Blockchain-related research that will impact e-learning? The authors answered to these questions by means a "systematic literature review" (terms used by authors). This is not a innovative or novelty approach, these type of papers have been published from 2018 to 2023, additionally it is limited to blockchain not includes all DLT architectures. However, these are not the unique weaknesses of the paper, here I found some other:
- A systematic literature review needs the definition of more exhaustive methodology, I mean, they need to define a general measurement or evaluation method to determine the impact of each reference, in order to identify the main authors, main journals, and main topics or technologies. There are not any summary reference table.
- Additionally, it is not clear whether the authors used the same methodology to projects and publications.
- Most of the references did not include publication year.
- The systematic literature review is incomplete. The literature review was limited to sciencedirect, scopus, and google schoolar, and IEEE, wiley, springer, etc.
- The discussion must be extended providing the conclusions provided by the application of proposed technology.
- Additionally, there are several platforms and consortium dedicated to the implementation of global blockchain, and some of then has projects in the e-learning scope.
- The section 4.4 try to show the future research trends related to blockchain, pretend to be the emerging trends, but the authors only described what is the summary content of some references, but they do not deep into the future trends or consequences of these references. In my point of view, the part of emerging trends is not included, of course some interesting aspects are analysed, but the authors did not provide the emerging trends, this type of information usually is the result of market study, new projects, in the future research lines of the references, etc.
- In discussion section, the section starts with: "our research focused on investigating the optimization of e-learning through advanced technologies like Blockchains" This does not fit with the title of the paper. The title is "Enhancing e-learning with blockchain": optimization is the only enhancement provided by Blockchain.
In conclusion, the proposed paper should be correctly focused and extended in order to provide a complete analysis of the scope of application of Blockchain in e-learning, and to answer to the proposed questions, in the current form is technically and methodologically incomplete.
Reviewer 3 Report
The article objectives are clear and well presented. It aims to assess the progress of Blockchain researches and projects in e-learning.The proposed methodology is well presented.
The work is well structured and English language is good. The article is written intelligibly in a scientific style and is easy to read. The title and abstract reflect very well to the content. The conclusion summarized the proposed work.
Тhe references are correct.
The figures are with good quality and readable.
Reviewer 4 Report
First of all the manuscript type is review not article.
3. Methodology, require introductory sentence before 3.1.
4. Results, require introductory sentence before 4.1.
Conclusion section is to long, make it short and take most to the discussion section. This will extend the discussion section to make it more appropriate.
Most of the references are not complete and are very old.
For a review paper 56 references in total is very less.
Consider to enhance by including recent and strong literature.
English language is fine. But authors shall double check to eliminate mistakes and typos.
Round 2
Reviewer 2 Report
Congratulations.
Reviewer 4 Report
All the comments are appropriately addressed.
English is good, better the authors doublecheck to avoid any mistakes and typos.